# PromptMix: A Class Boundary Augmentation Method for Large Language Model Distillation

**Gaurav Sahu**
University of Waterloo
gsahu@uwaterloo.ca

**Olga Vechtomova**
University of Waterloo

**Dzmitry Bahdanau**
ServiceNow Research
Mila, McGill University
Canada CIFAR AI Chair

**Issam H. Laradji**
ServiceNow Research

## Abstract

Data augmentation is a widely used technique to address the problem of text classification when there is a limited amount of training data. Recent work often tackles this problem using large language models (LLMs) like GPT3 that can generate new examples given already available ones. In this work, we propose a method to generate more helpful augmented data by utilizing the LLM's abilities to follow instructions and perform few-shot classifications. Our specific PromptMix method consists of two steps: **1)** generate challenging text augmentations near class boundaries; however, generating borderline examples increases the risk of false positives in the dataset, so we **2)** relabel the text augmentations using a prompting-based LLM classifier to enhance the correctness of labels in the generated data. We evaluate the proposed method in challenging 2-shot and zero-shot settings on four text classification datasets: Banking77, TREC6, Subjectivity (SUBJ), and Twitter Complaints. Our experiments show that generating and, crucially, relabeling borderline examples facilitates the transfer of knowledge of a massive LLM like GPT3.5-turbo into smaller and cheaper classifiers like DistilBERT$_{base}$ and BERT$_{base}$. Furthermore, 2-shot PromptMix outperforms multiple 5-shot data augmentation methods on the four datasets. Our code is available at https://github.com/ServiceNow/PromptMix-EMNLP-2023.

## 1 Introduction

Data scarcity is a key challenge in numerous real-life scenarios, such as deploying intent detection systems in task-oriented conversational agents and identifying hateful instances of speech on social media platforms. Crowdsourcing has been a traditionally popular choice to obtain additional data, but it is a financially expensive procedure incurring a high cost of human labor (Sheng et al., 2008;

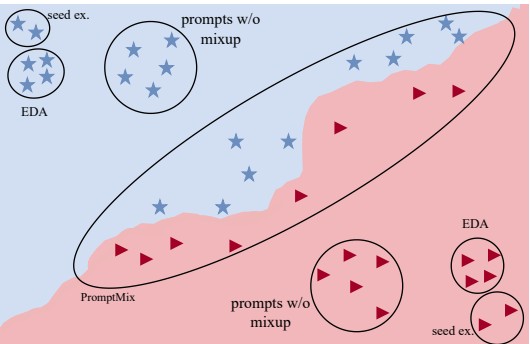

Figure 1: PromptMix focuses on generating examples near the class boundary of two classes, unlike other standard augmentation approaches like EDA (Wei and Zou, 2019) and prompting-based methods without Mixup (Sahu et al., 2022; Lin et al., 2023), that only use the information of a single class for augmentation.

Rashtchian et al., 2010; Rajpurkar et al., 2016; Khot et al., 2018). With the recent surge in the development of generative large language models (LLMs) (Brown et al., 2020; Chowdhery et al., 2022; Zhang et al., 2022; Touvron et al., 2023), a large body of literature has emerged that employs LLMs to generate additional data for various tasks (Kumar et al., 2020; Schick and Schütze, 2021; Wang et al., 2023).

In this work, we focus on the task of few-shot text classification (Schick and Schütze, 2021; Alex et al., 2021; Bragg et al., 2021). Specifically, we explore zero-shot and 2-shot settings. Early works employing LLMs to generate additional data samples for text classification first fine-tune a generative language model on an initial seed dataset and then use it to synthesize new training data (Wu et al., 2019; Kumar et al., 2019, 2020; Anaby-Tavor et al., 2020); however, the fine-tuning step quickly becomes a bottleneck in the absence of sufficient

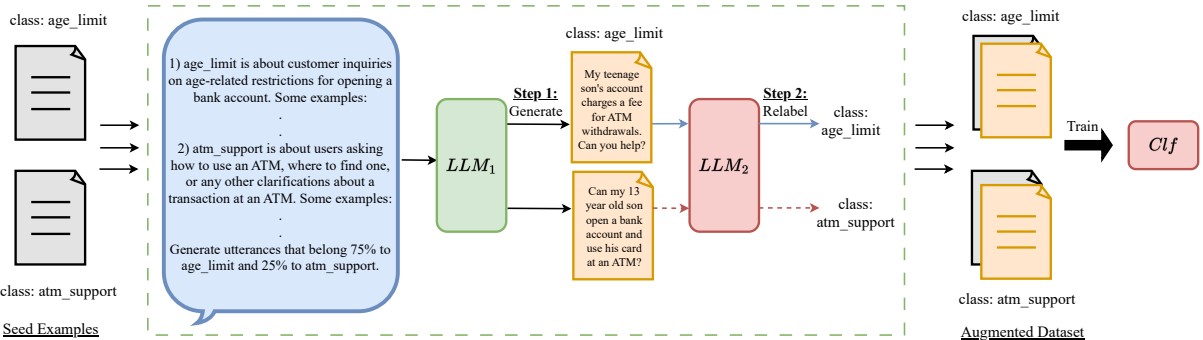

Figure 2: **PromptMix framework.** The dashed box shows the generation process for every class in the dataset. **Step 1:** we generate augmentations (yellow documents) by feeding the Mixup prompt proposed in Section 3.1 to an LLM. **Step 2:** we relabel _all_ the augmentations using an LLM (as described in Section 3.2) to fix any incorrect labels from Step 1. **Note:** $LLM_1$ and $LLM_2$ can be identical. Refer to Figures 3 and 4 for detailed versions of our prompts.

seed examples. More recent works sidestep fine-tuning by designing natural language prompts for off-the-shelf LLMs (Yoo et al., 2021; Sahu et al., 2022; Lin et al., 2023). A key limitation of such works is that their prompts only focus on using information from a single class when generating augmentations. Additionally, they do not incentivize the LLM to diversify the generated examples, and recent works show that instruction-tuned LLMs like InstructGPT (Ouyang et al., 2022) and Vicuna (Chiang et al., 2023) are prone to mode collapse (Zhu et al., 2023).

To address the previous limitations, we propose a two-step prompting-based technique, PromptMix. First, PromptMix instructs an LLM (in our case, GPT3.5-turbo[1]) to generate new examples by mixing information from multiple classes. The degree of mixing is controlled by a parameter $\alpha$, and using a range of $\alpha$ values diversifies the generated examples; however, promoting mixup increases the risk of false positive generations, so PromptMix uses a relabelling scheme in the second step to improve the faithfulness of the generated examples. In particular, it uses an LLM as a classifier to assign new labels to the generated examples. We find that training a classifier on these relabeled examples effectively transfers the knowledge of a massive LLM like GPT3.5 into much smaller models like BERT (Devlin et al., 2019) and DistilBERT (Sanh et al., 2019). Figure 2 demonstrates the complete PromptMix framework.

We summarize our contributions as follows: **a)** we propose PromptMix, a novel two-step prompting-based method to generate a diverse

set of labeled examples for any text classification dataset; and **b)** we demonstrate that generating borderline examples and relabeling them improves knowledge transfer from a massive LLM like GPT3.5 into much smaller models like DistilBERT and BERT, even without abundant seed examples. We also show that 2-shot PromptMix outperforms multiple data augmentation baselines that use 5-shot or more seed data.

## 2 Related work

Our work intersects with the topics of data augmentation, few-shot classification, and knowledge distillation, which we explain in detail below.

### 2.1 LLM-based Data Augmentation for Few-shot Text Classification

Kumar et al. (2019) evaluate different feature space data augmentation techniques, such as upsampling, perturbation, and linear interpolation, for varying levels of data scarcity (ranging from 5% data availability to 20%); however, their performance gains are minor compared to a no-augmentation setting. Kumar et al. (2020) consider 10-shot, 50-shot, and 100-shot text classification setups where they fine-tune pretrained language models like BERT, BART (Lewis et al., 2020), and GPT-2 (Radford et al., 2019) as data generators on $k-$shot data. Next, they condition the fine-tuned data generators to synthesize new training examples for individual classes in the dataset. This method improves over classical data augmentation techniques, such as easy data augmentation (Wei and Zou, 2019) and back translation (Sennrich et al., 2016), but the experiments were performed on datasets with very

---

[1]https://platform.openai.com/docs/models/gpt-3-5

few classes (up to seven). In reality, classification setups can have hundreds of classes, and the initial fine-tuning step would become a bottleneck.

To alleviate the fine-tuning bottleneck, Yoo et al. (2021) use natural language prompts for data augmentation; however, they also experiment with less challenging classification setups (with fewer classes). Sahu et al. (2022) propose a prompting-based approach using off-the-shelf GPT-3, to generate a large labeled corpus of augmented data. Their method improves across multiple classification setups with a large number of classes (up to 150) and varying levels of granularity; however, their method struggles on tasks where the classes carry very close semantic meanings. In particular, the generated samples have incorrect labels. In our method, we use a language model as a classifier to improve the label accuracy of the generated dataset.

Some non-prompting-based approaches include Wei et al. (2021), who propose curriculum data augmentation, where they first train a model on limited $k-$shot data and then incrementally introduce augmented data as training progresses. They use triplet loss (Schroff et al., 2015), which minimizes the distance between data points with the same label and maximizes for differently labeled examples. Tunstall et al. (2022) propose SetFit that first fine-tunes sentence transformers on a small number of text pairs in a contrastive Siamese manner (Dong and Shen, 2018) and then generate rich text embeddings for training a classification head. Finally, Kim et al. (2021) propose LINDA, which is first trained on one million sentence pairs randomly drawn from English Wikipedia. Later, they use the Mixup algorithm (Zhang et al., 2018) to interpolate between two English sentences of varying lengths. LINDA significantly improves the performance of a $\text{BERT}_{base}$ model across multiple few-shot classification setups. In our work, we consider more extreme few-shot setups (2-shot, zero-shot) and perform more controlled interpolation that promotes the generation of examples near class boundaries.

## 2.2 Knowledge Distillation

Knowledge distillation (Bucila et al., 2006; Hinton et al., 2015; West et al., 2022) refers to training a smaller student model mimicking the behavior of a much larger teacher model. In particular, the objective function aims to match the output distribution of the student model to that of the teacher model. By doing so, knowledge of the teacher model is effectively distilled into a much smaller student model, allowing a similar level of performance as the teacher at a lower computational cost. Shridhar et al. (2022) distill a GPT3 (6B) model into a GPT-2 model for a Chain-of-thought (CoT) reasoning task. Liang et al. (2021) propose MixKD to encourage the student model to mimic the teacher's behavior on not only the available training examples but also on interpolated examples. Finally, Sun et al. (2020) distill knowledge through intermediate layers of the teacher via a contrastive objective. In comparison, our approach allows knowledge distillation of a massive teacher model like GPT3.5 (175B parameters) into significantly smaller student models like DistilBERT and BERT (67M and 110M parameters, respectively).

## 3 Methodology

We hypothesize that training a robust text classifier requires the training data to have a good mix of borderline examples (Swayamdipta et al., 2020). This section describes our method PromptMix, where we first instruct an LLM (GPT3.5-turbo, in our case) to generate difficult examples near class boundaries then relabel those generations to improve the faithfulness of the generated data (see Figure 2).

## 3.1 Step 1: Generating examples

**First**, we manually write short descriptions for every class in the dataset. We use descriptions in our prompts to facilitate the usage of the approach in the extremely data-scarce zero-shot and two-shot settings. **Next**, we randomly select a group of $t(=4)$ [2] classes $c \subseteq C$ classes, where $C$ denotes the set of all classes in the dataset. For each class in $c$, we combine the description and $k$ examples in the prompt, $k$ being the $k-$shot setting (see part 1 in Figure 3). **Lastly**, for each class $c_i \forall i \in [1, t]$ in the subset, we instruct GPT3.5-turbo [3] to generate $n$ example utterances that are a mix of two classes: $c_i$ and a randomly selected class $c_j \in c \setminus \{c_i\}$. In particular, we instruct the LLM to generate utterances that belong $\alpha\%$ to class $c_i$ and $(1 - \alpha)\%$ to class $c_j$ (see part 2 in Figure 3). Figure 5 in Appedix A.1 shows the distribution $\alpha$ is sampled from.

---

[2] We choose $t = 4$ based on the results in Section A.2
[3] specifically, we use the `gpt-3.5-turbo-0613` engine

```
Input Prompt:
Consider the task of classifying between the following
classes(along with some examples):

1. age_limit, which is about customer inquiries on
age-related restrictions for opening a bank account.
Some examples of utterances include:
- Can I get an account for my son?
- Can my teenager have an account?

2. atm_support, which is about users asking how to
use an ATM, where to find one, or any other
clarifications about a transaction at an ATM. Some
examples of utterances include:
- Is the closest ATM to me within 2 miles?
- Are there only certain ATM machines where I can
  use this card?                                    } Part 1

Generate a diverse set of 4 short utterances where
each utterance belongs 75% to age_limit and 25% to  } Part 2
atm_support.

Example 1:

Completions:

- Can someone under 18 open an account with unlimited
  ATM withdrawal limit?
- Can I open an account for my teenage daughter?
- Do I need to be over a certain age to use an ATM?
- Can I use my children's debit card to withdraw money
  from the ATM?
```

Figure 3: **PromptMix prompt.** The demonstration highlights the two main parts of our prompt: Part 1 shows the description and examples, and Part 2 shows the mixup instruction. We use GPT3.5-turbo to obtain the completions. In this example, we highlight good generations in blue and bad generations in red. **Note:** for brevity, we only include two classes in the prompt.

```
Input Prompt:
Consider the task of classifying between the following
classes(along with some examples):

1. age_limit, which is about customer inquiries on
age-related restrictions for opening a bank account.
Some examples of utterances include:
- Can I get an account for my son?
- Can my teenager have an account?

2. atm_support, which is about users asking how to
use an ATM, where to find one, or any other
clarifications about a transaction at an ATM. Some
examples of utterances include:
- Is the closest ATM to me within 2 miles?
- Are there only certain ATM machines where I can
  use this card?

Consider the following test sentence:
1. Do I need to be over a certain age to use an ATM?

Classify the test sentence into one of the previously
described classes.

1.

Example 1:

Completions:

- atm_support
```

Figure 4: **Relabelling prompt.** The generated sentence does not belong to the class age_limit as the main objective of the query entails an ATM support query, which is fixed after relabeling. **Note:** we relabel both good and bad examples. The color in the figure is just for demonstration purposes.

## 3.2 Step 2: Improving Fidelity

Since borderline examples are inherently difficult to put into a category, the LLM may generate examples for the minority class in the prompt ($c_j$). In other words, the LLM can generate false positives. To address this issue, we employ GPT3.5-turbo as a classifier and relabel the generated examples. When constructing the classification prompt, we choose the top-5 closest classes according to the similarity between the SBERT sentence embedding (Reimers and Gurevych, 2019) of the generated example and available examples in the few-shot training set. For the zero-shot setting, we use the class name's embedding instead of the available examples. We then follow a similar process as in Step 1 to construct the prompt, but instead ask the LLM to classify the provided sentence into one of the classes in the prompt (see Figure 4). To ensure a valid prediction, we retrieve the closest class in the dataset based on the cosine similarity of the SBERT embedding of the GPT-generated class and the ground-truth classes in the dataset [4]. We do

---

[4] we use the `sentence-transformers/all-mpnet-base-v2` model from the sentence-transformers library

not include all the classes in the prompt because **a)** some datasets can have hundreds of classes that would not fit in the context size of the LLM, and **b)** we found in our preliminary experiments that long contexts degraded GPT's classification ability. Figure 4 shows the prompt structure for relabeling a generated example.

After generating borderline examples and relabeling them, we train a text classifier on the combined PromptMix-generated data and the original seed data. Specifically, we fine-tune DistilBERT$_{base}$ and BERT$_{base}$ classification models. In the subsequent sections, we show that our method achieves text augmentation, pseudo-labeling, and knowledge distillation in one cohesive pipeline.

## 4 Does PromptMix Generate Borderline Examples?

Table 1 shows that using mixup generates sentences that contain information from the majority and the minority class, compared to sentences generated without mixup that only contain information about the majority class. This demonstrates that mixup

| | |
|---|---|
| Input classes | age_limit (75%) atm_support (25%) |
| w/ Mixup | - Can someone under 18 open an account with unlimited ATM withdrawal limit? 
 - Can I open an account for my teenage daughter? 
 - Do I need to be over a certain age to use an ATM? 
 - Can I use my children's debit card to withdraw money from the ATM? |
| w/o Mixup | - Can I open an account for a minor? 
 - What is the minimum age requirement to get a bank account? 
 - Is there an age restriction for account holders? 
 - Is there a minimum age requirement to open an account? |

Table 1: **Effect of Mixup.** GPT3.5-turbo generations for the prompt shown in Figure 3. We've highlighted parts about age_limit in yellow and parts about atm_support in cyan. We see clear evidence of mixup at work as sentences generated using mixup contain information about both classes.

| | B77 | TREC6 | SUBJ | TC |
|---|---|---|---|---|
| Classes | 77 | 6 | 2 | 2 |
| # Train | 9002* | 5452 | 8000 | 100* |
| # Valid | 1001* | 500 | 2000 | - |
| # Test | 3080 | 500 | 2000 | 3349* |

Table 2: Statistics of the text classification datasets we use in our experiments. * indicates that we split the original data into training and validation/testing instead of using a split provided by the dataset authors.

leads to the generation of borderline examples.

Table 1 also shows some false positives, where GPT generates sentences like, "Do I need to be over a certain age to use an ATM?" for the majority class age_limit. The class age_limit covers all age-related queries about *opening a bank account*, whereas the generated sentence is an age-related query about *using an ATM*, making it a better fit for the minority class atm_support. By relabelling such false positives, we aim to rectify the mismatch between the generated examples and their assigned class labels. Figure 4 shows that GPT correctly predicts the new class of the sentence as atm_support, verifying the importance of the relabeling step in our approach.

Overall, we find the observations in this section to be strong evidence that PromptMix can generate high-quality datasets even in aggressive few-shot text classification setups. Next, we conduct an extensive suite of experiments to verify the effectiveness of PromptMix.

# 5 Experimental Setup

## 5.1 Datasets

We use four text classification datasets with varying levels of granularity among the classes. Banking77 (**B77**) (Casanueva et al., 2020) is a single-domain dataset with 77 banking-related classes, where the difference between multiple classes is nuanced. The fine-grained nature combined with a high number of target classes makes Banking77 a good test

bed for verifying the scalability and effectiveness of our approach. The following three datasets have coarse labels but cover a variety of domains, allowing us to test the adaptability of our method across different domains. **TREC6** (Voorhees et al., 1999) is a question classification dataset with six broad classes of questions in English. The subjectivity dataset (**SUBJ**) (Pang and Lee, 2004) contains movie reviews with objectivity labels. Lastly, the twitter complaints dataset (**TC**) (Preoţiuc-Pietro et al., 2019) contains tweets annotated by whether they contain a complaint or not. We refer the reader to Table 2 for exact statistics of all the datasets.

## 5.2 Few-shot Setup

For our experiments, we consider 1) a 2-shot setup, where only $k = 2$ training examples are available for every class, and 2) a zero-shot setup, where we do not have access to any training examples.

**Notations.** We will use $D_{part}$ to refer to the dataset parts, i.e., train, validation, and test. When augmenting the training data using any method, we generate $N$ examples per class and refer to the resulting data as $\tilde{D}_{A,train}$ (obtained after Step 1). We refer to the relabeled version of the resulting data (obtained after Step 2) as $\tilde{D}_{A+R,train}$.

## 5.3 Training and Evaluation

**Training.** We fine-tune DistilBERT$_{base}$ and BERT$_{base}$ models for text classification by adding a linear layer on top of the [CLS] token (Wolf et al., 2019). In all the setups, we fine-tune the classifier for 5 epochs. We use a learning rate of $6 \times 10^{-5}$ and weight decay of $1 \times 10^{-3}$ for B77 and a learning rate of $4 \times 10^{-5}$ and weight decay of $1 \times 10^{-2}$ for all the other datasets. Finally, we generate $N = 50$ examples per class for B77 and TREC6 and $N = 100$ examples per class for SUBJ and TC. We used the validation sets of TREC6 and SUBJ to choose the specific $N$ values as they provide a good cost-to-performance ratio. We chose TREC6 over B77 to

minimize our costs for using GPT3.5-turbo.

We perform all hyperparameter tuning using DistilBERT$_{base}$ on B77 and TREC6. We limit our tuning to the two datasets to obtain two sets of hyperparameters: one for large-scale datasets like B77 and another for small-scale datasets like TREC6, SUBJ, and TC. Additionally, we use the same set of hyperparameters for the BERT$_{base}$ classification model. We use the full validation set for tuning instead of a few-shot one to avoid issues with unstable hyperparameters.

We run experiments for the following scenarios: **1) Baseline (2-shot).** All the classes are reduced to 2 examples per class, and we fine-tune a DistilBERT$_{base}$/BERT$_{base}$ model on the reduced dataset. **2) NN+GPT3.5.** We use the nearest-neighbor approach to populate the prompt as described in Section 3.2 and then prompt GPT3.5-turbo to classify the test set examples (see Figure 4 for reference). **3) Sahu et al. (2022).** A prompting-based approach for data augmentation that lists all the seed examples for a single class in the prompt and prompts the LLM to generate more examples based on it. It does not promote mixup in the generated examples. **4) PromptMix.** Our proposed approach with multiple classes in the prompt, instructing the LLM to generate borderline examples. **5) PromptMix (zero-shot).** We remove seed examples from PromptMix but still use the manually written class descriptions. **6) Easy Data Augmentation (EDA).** An edit-based augmentation technique proposed by Wei and Zou (2019) that applies rule-based changes to existing training examples to generate additional examples. **7) GPT3Mix.** A mixup-based augmentation method using soft labels for pseudolabeling proposed by Yoo et al. (2021). Notably, Yoo et al. (2021) measure the degree of few-shot in terms of the percentage of training examples used for augmentation (sub-sample percentage) and experiment with different percentages. We report their best-performing model for a sub-sample percentage of 1%, where GPT3Mix uses 55 training examples in TREC6 and 80 training examples in SUBJ (combined for all the classes). **8) CPFT.** Zhang et al. (2021) use contrastive pre-training before fine-tuning a model for few-shot intent detection using RoBERTa$_{base}$. **9) USE.** A large multilingual model pretrained on 16 languages (Yang et al., 2020). **10) CONVERT.** An intent detection model from Casanueva et al. (2020), which uses a dual encoder setup pretrained

on 654 million Reddit sentences.

Several works in the past have explored 5-shot, 10-shot, and even 100-shot settings for data augmentation (Kumar et al., 2020; Zhang et al., 2021). But, to the best of our knowledge, we are the first to explore data augmentation in 2-shot and zero-shot settings. Through our experiments, we aim to measure the extent of data scarcity that an LLM like GPT3.5-turbo can handle, which is optimized to follow instructions.

**Evaluation.** We measure the performance of the classifiers in terms of test classification accuracy and report the full set of results in Table 3. To note, **A1** denotes the accuracy of the classifier on $D_{train} \cup D_{A,train}$ (augmented dataset after Step 1 in Section 3.1) and **A2** denotes the accuracy of the classifier on $D_{train} \cup D_{A+R,train}$ (augmented+relabeled dataset after Step 2 in Section 3.2). We run each experiment for three random seeds and report the mean accuracy in Table 3.

## 6 Results

Referring to Table 3, we first note that A1 performs significantly better than baseline (2-shot) for DistilBERT$_{base}$ and BERT$_{base}$ across all four datasets. This confirms that data augmentation is helpful in data-scarce setups.

### 6.1 On the Effect of Relabeling Step

Table 5 shows the percentage of generated examples relabeled by GPT3.5-turbo for different augmentation methods. We note that relabeling percentage is higher when we add mixup to the prompt (PromptMix v/s PromptMix w/o Mixup). High relabeling percentages suggest that mixup generates more borderline examples than any other augmentation method. This verifies the importance of the relabeling step in our method, which is highly effective in remedying the problem of false positive generations. This is further demonstrated by higher A2 values than A1 across the board. Specifically, for PromptMix, we observe an improvement of 7.4% on B77, 8.1% on TREC6, 12.4% on SUBJ, and 13.5% on TC, when we use DistilBERT$_{base}$; and 5.9% on B77, 10.4% on TREC6, 14.4% on SUBJ, and 11.6% on TC when we use BERT$_{base}$. In addition to these significant improvements, we note that the degree of improvement increases as the classification task gets easier in terms of the number of target classes. Table 6 shows some examples of leaked generations that GPT rectifies

| Method | Prompt Features | | | | B77 | | TREC6 | | SUBJ | | TC | |
|---|---|---|---|---|---|---|---|---|---|---|---|---|
| | Ex. | Desc. | >1 class | Mixup | A1 | A2 | A1 | A2 | A1 | A2 | A1 | A2 |
| **DistilBERT$_{base}$** | | | | | | | | | | | | |
| Baseline | - | - | - | - | 16.0 (0.9) | | 31.7 (0.8) | | 64.4 (0.7) | | 38.8 (0.6) | |
| EDA | - | - | - | - | 47.8 (0.7) | | 40.9 (0.6) | | 82.3 (0.4) | | 42.9 (0.7) | |
| GPT3Mix | ✓ | | ✓ | ✓ | - | | 57.4 (2.8) | | 89.3 (1.5) | | - | |
| Sahu et al. (2022) | ✓ | | | | 68.9 (1.4) | 71.6 (0.6) | 51.1 (1.3) | 56.9 (0.7) | 81.8 (1.3) | 83.7 (0.4) | 51.5 (0.6) | 55.4 (0.5) |
| + desc. | ✓ | ✓ | | | 71.1 (1.2) | 72.4 (0.7) | 63.8 (1.1) | 64.9 (0.6) | 84.2 (1.1) | 86.3 (0.5) | 57.2 (0.7) | 67.9 (0.2) |
| PromptMix w/o Mixup (zero-shot) | | ✓ | ✓ | | 72.2 (1.3) | 76.1 (0.8) | 61.0 (1.3) | 61.6 (0.6) | 82.5 (1.2) | 84.2 (0.5) | 56.7 (1.3) | 67.7 (0.5) |
| PromptMix (zero-shot) | | ✓ | ✓ | ✓ | 69.2 (2.3) | 77.4 (1.2) | 54.1 (1.7) | 65.7 (0.7) | 80.0 (1.5) | 85.4 (0.6) | 56.0 (1.3) | 71.5 (0.9) |
| PromptMix w/o Mixup | ✓ | ✓ | ✓ | | 73.1 (1.3) | 78.4 (0.5) | 65.4 (1.2) | 66.2 (0.5) | 85.4 (1.2) | 87.8 (0.3) | 64.6 (0.7) | 71.5 (0.6) |
| PromptMix | ✓ | ✓ | ✓ | ✓ | 72.3 (1.1) | **79.7 (0.7)** | 60.6 (1.4) | **68.7 (0.6)** | 77.5 (1.7) | **89.9 (0.8)** | 61.8 (1.4) | **75.3 (1.2)** |
| **BERT$_{base}$** | | | | | | | | | | | | |
| Baseline | - | - | - | - | 22.6 (1.2) | | 33.0 (0.6) | | 71.6 (0.8) | | 42.7 (0.6) | |
| EDA | - | - | - | - | 49.2 (0.9) | | 51.1 (0.6) | | 84.5 (0.5) | | 47.8 (0.4) | |
| GPT3Mix | ✓ | | ✓ | ✓ | - | | 60.5 (6.1) | | 90.6 (1.1) | | - | |
| LINDA (5-shot) | ✓ | | ✓ | ✓ | - | | 62.2 (3.1) | | - | | - | |
| Sahu et al. (2022) | ✓ | | | | 70.7 (1.4) | 71.6 (1.1) | 51.3 (1.2) | 57.8 (0.8) | 83.6 (1.3) | 85.5 (1.0) | 55.3 (1.1) | 58.1 (0.3) |
| + desc. | ✓ | ✓ | | | 72.8 (1.1) | 73.0 (0.7) | 64.3 (1.3) | 67.1 (0.2) | 87.0 (1.7) | 87.2 (1.1) | 66.1 (1.4) | 65.2 (1.1) |
| PromptMix w/o Mixup (zero-shot) | | ✓ | ✓ | | 74.0 (1.9) | 76.4 (1.1) | 64.2 (1.6) | 68.5 (0.9) | 83.6 (1.4) | 85.9 (0.8) | 64.3 (1.2) | 68.9 (0.3) |
| PromptMix (zero-shot) | | ✓ | ✓ | ✓ | 71.3 (2.4) | 77.6 (1.3) | 55.5 (1.7) | 67.5 (0.4) | 80.7 (1.4) | 89.5 (0.7) | 57.6 (2.4) | 74.7 (1.5) |
| PromptMix w/o Mixup | ✓ | ✓ | ✓ | | 74.4 (1.4) | 78.5 (0.7) | 70.1 (1.5) | 71.6 (0.2) | 87.0 (1.3) | 90.0 (0.1) | 71.0 (0.9) | 72.3 (0.3) |
| PromptMix | ✓ | ✓ | ✓ | ✓ | 74.2 (1.6) | **80.1 (0.9)** | 63.3 (2.5) | **73.7 (1.1)** | 77.3 (2.2) | **91.7 (1.1)** | 66.8 (0.8) | **78.4 (0.8)** |
| NN+GPT3.5 | ✓ | ✓ | ✓ | - | 79.9 | | 74.4 | | 90.4 | | 88.6 | |

Table 3: Test classification accuracy (out of $100\%$) on four datasets, averaged across three random seeds (with standard deviation in brackets). A1 is the accuracy of the classifier on the generated dataset, and A2 is the accuracy on the generated+relabeled dataset. **Note:** we use GPT3Mix results from Yoo et al. (2021), and LINDA results from Kim et al. (2021). "-" for a method shows that the particular prompt feature is inapplicable to that method.

| Method | Accuracy |
|---|---|
| USE (5-shot) | 76.3 |
| CONVERT (5-shot) | 75.3 |
| USE+CONVERT (5-shot) | 77.8 |
| CPFT (5-shot) | **80.9** |
| PromptMix (2-shot) | 80.1 |

Table 4: Comparing 2-shot PromptMix results with 5-shot baselines on B77. **Note:** CPFT, USE, CONVERT, and USE+CONVERT as reported in Zhang et al. (2021).

| Method | B77 | TREC6 | SUBJ | TC |
|---|---|---|---|---|
| Sahu et al. (2022) | 9.4 | 21.0 | 3.9 | 2.5 |
| + desc. | 8.5 | 28.0 | 1.5 | 8.5 |
| PromptMix (zero-shot) | 32.8 | 36.2 | 28.8 | 16.2 |
| w/o Mixup | 22.5 | 18.9 | 2.6 | 7.8 |
| PromptMix | **33.9** | **33.8** | **42.0** | **23.4** |
| w/o Mixup | 22.2 | 20.3 | 6.1 | 14.0 |

Table 5: Percentage of generated examples relabeled by GPT3.5-turbo in Step 2 for different methods. **Note:** percentages are averaged across three runs.

during the relabeling step.

## 6.2 Borderline Examples Aid Knowledge Distillation

In Table 3, we notice that PromptMix achieves almost similar performance as NN+GPT3.5 on three out of four datasets. Specifically, Prompt-Mix outperforms NN+GPT3.5 on B77 (80.1 v/s 79.9) and SUBJ (91.7 v/s 90.4) and is competitive on TREC6 (73.7 v/s 74.4). The gap between PromptMix and NN+GPT3.5 is larger on TC compared to other datasets (78.4 v/s 88.6). This might be due to the nature of the dataset, which contains extensive use of social media language that GPT knows about, but two examples might be too few to cover the vast diversity of complaints in the wild. Overall, these results are promising as GPT3.5-turbo with 175B parameters is $\sim 2600\times$ larger than DistilBERT$_{base}$ and $\sim 1600\times$ larger than BERT$_{base}$, which only have 67M and 110M parameters, respectively. In both cases, our final classifiers are $> 99.9\%$ smaller than GPT3.5-turbo. We do not see such strong performance for any other generation method, even after relabeling. This confirms that generating borderline examples in PromptMix makes for a high-quality dataset generation method that aids the transfer of knowledge of large-scale models such as GPT3.5-turbo into much smaller classifiers in data-scarce settings.

## 6.3 PromptMix v/s Other Augmentation Methods

In Table 3, we compare the 2-shot performance of PromptMix against several strong baselines on the four datasets. First, we note that PromptMix is significantly better than the EDA baseline on all datasets. Next, we compare the results of Prompt-Mix (which uses GPT3.5-turbo for generation) with

| GPT3.5-turbo generated sentences | Before Relabeling → After Relabeling |
|---|---|
| **Twitter Complaints** | |
| This tweet also expresses an opinion about a product, but includes a minor complaint about the battery life. | no_complaint → complaint |
| I appreciate the quick response from @Uber_Support, but the driver's behavior was unacceptable. #goodandbad | no_complaint → complaint |
| @Starbucks this latte is amazing! Can you tell me what kind of beans you use? #No_Complaint | complaint → no_complaint |
| I'm really impressed with @Nike's customer service. They helped me solve my problem quickly and efficiently. | complaint → no_complaint |
| **Banking77** | |
| Why was my transfer declined after I tried to send money to my daughter's account? | age_limit → declined_transfer |
| My transfers keep getting declined, but I'm old enough to have an account. Can you help me? | age_limit → declined_transfer |
| My card got declined when I tried to top up my account. What's going on? | topping_up_by_card → declined_card_payment |
| Is there a limit to how much I can top up with my card? It's not letting me add more money. | topping_up_by_card → top_up_limits |
| I just tried topping up my account with my card and it didn't work. | topping_up_by_card → top_up_failed |

Table 6: **Effect of Relabeling.** GPT3.5-turbo sentences that were relabeled in the Twitter Complaints and Banking77 datasets. Highlighted text denotes the difference with respect to the class of the generated sentence before relabeling.

GPT3Mix (which uses GPT3's Davinci model for generation) and LINDA as they are the closest methods to ours in terms of motivation. Prompt-Mix outperforms both LINDA and GPT3Mix by a huge margin even though LINDA is 5-shot and uses pretraining, and GPT3Mix utilizes 1% of training samples for augmentation, translating to 55 total seed examples compared to our 12 seed examples on TREC6, and 80 seed examples compared to our 4 seed examples on the SUBJ dataset.

Table 4 shows that 2-shot PromptMix outperforms USE, CONVERT, and USE+CONVERT on the Banking77 dataset. It also achieves an accuracy of 80.1% compared to 80.9% by CPFT, which is highly competitive. To emphasize, all the considered baselines are 5-shot, with some models using additional data for pretraining. Moreover, it is possible to further improve the performance of PromptMix with more dataset- and model- specific hyperparameter-tuning; however, our core message is to show that generating borderline examples combined with relabeling is a highly promising method for data augmentation in 2-shot and zero-shot text classification setups.

### 6.4 Descriptions are Impactful

Our experiments that were used to evaluate the effectiveness of using human-written descriptions show promising results. First, we note that simply adding a class description to the prompt leads to decent performance gains on all the datasets (Sahu et al. (2022) w/ and w/o desc). Comparing Prompt-Mix (zero-shot) with and without Mixup against Sahu et al. (2022) + desc. shows that using only descriptions in the prompt (no seed examples) leads to a significant boost in model performance. We also observe that using multiple class descriptions in a prompt is either better or equivalent to using a single class with its description and examples. This suggests that providing more context about other classes helps LLM generate better quality augmentations for a given class. Therefore, it follows intuitively that PromptMix, which combines the usage of multiple classes with descriptions as well as seed examples, leads to the best performance on all the datasets.

### 6.5 Open-source v/s Closed-source models

Table 3 and 4 showcase strong capabilities of GPT3.5-turbo for our task. However, GPT3.5-turbo is a closed-source language model provided by OpenAI, and it can quickly get costly as we increase the number of classes in the dataset. Therefore, we conduct a small-scale experiment on the TREC6 dataset with open-source LLMs. In particular, we prompt open-source LLMs instead of GPT3.5-turbo to synthesize new examples and to relabel them. Next, we finetune a BERT$_{base}$ classi-

| Method | Accuracy |
|---|---|
| Baseline | 33.0 |
| PromptMix (Llama-2-7b-chat-hf) | 55.6 |
| PromptMix (Llama-2-13b-chat-hf) | 66.6 |
| PromptMix (Llama-2-70b-chat-hf) | 70.8 |
| PromptMix (GPT3.5-turbo) | 73.7 |
| PromptMix (GPT-4) | **86.2** |
| NN+GPT3.5 | 74.4 |
| NN+Llama-2-70b-chat-hf | 76.2 |
| NN+GPT4 | 88.2 |

Table 7: Comparison of different open-source LLMs v/s GPT on TREC6. **Note:** Baseline refers to the 2-shot accuracy of the BERT$_{base}$ classifier. We replace GPT3.5-turbo with GPT4 and LLama-2-70b-chat-hf to get the NN+GPT4 and NN+LLama-2-70b-chat-hf baselines, respectively.

fier on the augmented+relabeled dataset.

GPT-Neo (1.3B), GPT-J (6B), and the more recent instruction-tuned Vicuna and Stable-vicuna (13B) (Chiang et al., 2023) models achieve very poor performance compared to GPT3.5-turbo. We find that even for medium-sized LLMs like Vicuna and Stable-vicuna, inference time is a major bottleneck [5]. However, our experiments with LLama-2 show promising results. We used different-sized Llama-2 models (7b, 13b, 70b) on the TREC6 dataset and observe a strong correlation between model size and test accuracy. In particular, we note that LLama-2-70b-chat-hf might be a decent alternative to the closed-source GPT3.5-turbo model. We also test GPT-4 (larger than the GPT3.5 model) and observe a significant boost in classification accuracy. Table 7 shows the results [6].

## 7    Conclusion

To conclude, we propose PromptMix, a novel two-step prompting-based method for generating borderline augmented examples in extreme few-shot text classification setups. Our method combines the process of text augmentation, pseudolabeling, and knowledge distillation in a cohesive pipeline. We show that by generating borderline training examples and relabeling them using a large teacher model like GPT3.5-turbo, we can transfer the knowledge of such massive LLMs into much smaller models like DistilBERT$_{base}$ and BERT$_{base}$. Furthermore, 2-shot PromptMix beats multiple 5-shot or higher data augmentation baselines, making

it a highly promising data augmentation approach.

## Limitations

Our results indicate a promising potential to use LLMs for generating data in highly-aggressive few-shot setups; however, this work has a few limitations, as detailed in this section. First, we rely completely on GPT's pseudolabels to tackle false positive generations during the augmentation step. Secondly, since we use SBERT embeddings to ensure that pseudolabel is a valid class in the dataset, we ignore potential out-of-scope/out-of-domain (OOS/OOD) generations. We did observe a few instances where GPT pseudolabels were of the form, "This sentence does not belong to any of the provided classes." This presents a good avenue to introduce human interventions that can judge GPT pseudolabels and can identify OOS/OOD examples, which can be helpful in many real-life tasks. We also want to emphasize that while human interventions can greatly help, they also present the challenge of minimizing human labor.

Second, while our experiments with open-source models suggest that the larger open-source models like LLama-2-70b might be a good alternative to closed-source models like GPT3.5, thereby greatly cutting API costs, they still demand significant computational resources. Therefore, efforts towards distilling the knowledge of bigger LLMs like Llama-2-70b into smaller models would greatl aid in making these models more accessible and feasible for use.

## Ethics Statement

We use GPT to generate new examples, and even though GPT3.5-turbo is instruction-tuned, some generations might depict undesirable biases with respect to the current objective. For the stated reason, we recommend using our proposed classification models with human monitoring to avoid any ethical issues due to false positive predictions of the downstream classifiers. Practitioners may also consider explicitly debiasing language models for their specific use cases (Barikeri et al., 2021; Schick et al., 2021).

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

# A Appendix

## A.1 Distribution of $\alpha$ for Mixup

We use $\alpha = round(10(x+1))/20 \; \forall \; x \sim \beta(5,2)$ in our experiments. We modify the standard $\beta-$distribution by restricting its range to the half interval of $(0.5, 1.0]$ with a peak near $1.0$. We choose $\alpha > 0.5$ to incentivize the LLM to generate examples that are mixed up but still belong to $c_i$. We round off the $\alpha$ to the nearest $0.05$ to avoid decimal values that would be arbitrary, given we are operating with natural language instructions.

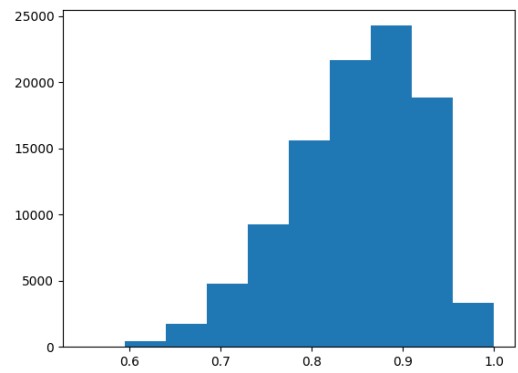

Figure 5: $\alpha = round(10(x+1))/20 \; \forall \; x \sim \beta(5,2)$.

## A.2 Choice of $t$

We experiment with different values of $t$ (classes to include in the prompt) and choose $t = 4$ based on the validation performance in Table 8.

| t | Accuracy |
|---|----------|
| 2 | 73.3 |
| 4 | 76.8 |
| 6 | 69.1 |

Table 8: Validation accuracy for different values of $t$ on TREC6.