# OpenReview forum: "PromptMix: A Class Boundary Augmentation Method for Large Language Model Distillation"
_EMNLP/2023/Conference — EMNLP 2023 Main_

### Official Review · Reviewer_ivSt · 2023-07-27

**Typos Grammar Style And Presentation Improvements:** N/A
**Soundness:** 4

**Excitement:**

3: Ambivalent: It has merits (e.g., it reports state-of-the-art results, the idea is nice), but there are key weaknesses (e.g., it describes incremental work), and it can significantly benefit from another round of revision. However, I won't object to accepting it if my co-reviewers champion it.

**Missing References:**

N/A

**Paper Topic And Main Contributions:**

The paper proposes a data augmentation approach (PromptMix) for generating additional training data for text classification that utlizes the abilities of large language models to follow instructions and perform few shot classification. The authors hypothesise that training a robust text classifier requires the training data to have a good mix of borderline examples. Therefore, they propose a two-step prompting-based method which first instructs GPT3.5-turbo to generate new examples by mixing information from multiple classes in order to produce borderline examples. Then, in the second step, the method uses a relabling scheme based on GPT3.5-turbo but this time instructing the model to relabel those generations to improve reliability of the class labels. When constructing the classification prompt, the authors select the top-5 closest classes according to the similarity between the SBERT sentence embedding.  The authors evaluate their method accross four diverse text classification datasets. The authors focus on 2-shot and zero-shot cases and fine-tune DistilBERT-base and BERT-base models for performing text classification. The results show that borderline examples aid knowledge distillation and show that 2-shot PromptMix beats multiple 5-shot or higher data augmentation baselines.

**Questions For The Authors:**

1) Did you follow any specific methodology for writing the prompts and the class descriptions?

**Reasons To Accept:**

The approach is very interesting and shows to work without the need of additional unlabelled data or fine-tuning the generative model. The novelty of the paper is well justified and shows that generating class border samples and relabling them using the same model lead to improvements over baseline methods. The authors have performed extensive analysis.

**Reasons To Reject:**

The OPEN AI models work quite well in few and zero shot settings using only prompts, however these services are paid. It could have been interesting to see how well the approach work with open source models which are much more sensitive to the prompts as well as the classification dataset used for the task. The authors do not discuss the methodology they used for writing the instructions or how dependent the model is to the dataset and the classes used for the task. It is not discussed how easy and whether is possible to adapt the approach simply by using prompts to diverse set of datasets such as legal domain, medical domain.

**Reproducibility:**

3: Could reproduce the results with some difficulty. The settings of parameters are underspecified or subjectively determined; the training/evaluation data are not widely available.

**Reviewer Confidence:**

4: Quite sure. I tried to check the important points carefully. It's unlikely, though conceivable, that I missed something that should affect my ratings.

---

> ### Author Rebuttal · Authors · 2023-08-28
>
> Dear Reviewer,
>
> Thank you for your detailed comments on our paper. We are glad you find our approach interesting and are satisfied with our analysis. You raise an excellent point about the feasibility of open-source models in our framework. We've addressed that comment and others in detail below. We would also like to mention that we will release our codebase upon acceptance for reproducibility.
>
> > *It could have been interesting to see how well the approach work with open source models which are much more sensitive to the prompts as well as the classification dataset used for the task.*
>
> Great question! We have tried the open-source models GPT-Neo (1.3B), GPT-J (6B), and the more recent instruction-tuned Vicuna and Stable Vicuna (13B) models but they achieved low performance in comparison to GPT-3.5.
>
> However, new open-source models have emerged recently in the LLM space, like Llama-2. As seen in the Table below, we ran different sized Llama-V2 (7b, 13b, and 70b) on TREC 6 and saw that Llama-V2 70b achieves close results to GPT-3.5, suggesting that current open-source models are also feasible for our proposed method. We have added these results to the paper.
>
>
> **Exp 1. Results on TREC 6**
>
> | Model |  Test Acc (A2) |
> | --- | --- |
> |**PromptMix Results**|
> | Llama-2-7b-chat-hf | 55.6$^\dagger$ |
> | Llama-2-13b-chat-hf |  66.6$^\dagger$ |
> | Llama-2-70b-chat-hf |  70.8$^\dagger$ |
> | GPT-3.5-turbo (as in the paper)  |  73.7 |
> |**Upper Bound Results**|
> |NN+GPT-3.5-turbo (as in the paper)  |  74.4 |
> ||
>
> $^\dagger$ used Llama-70b-chat-hf for relabeling.
>
> > *It is not discussed how easy and whether is possible to adapt the approach simply by using prompts to diverse set of datasets such as legal domain, medical domain.*
>
> Thanks for raising this point. For each domain we had around 3-4 iterations of writing the class descriptions. We chose the descriptions that gave us the most reasonable synthetically generated sentences on a very small set of examples, which makes it practical in the sense that it requires little human effort. Since this approach worked for a variety of diverse domains from banking to social media, we believe that this method will work on domains like legal and medical. We have added this detail in the paper.
>
> > *Did you follow any specific methodology for writing the prompts and the class descriptions?*
>
> We tried multiple class descriptions starting with the most general definitions of the classes and then finetuning them around 3-4 times (manually) as described in the previous answer.
>
> For constructing the prompts, we chose the one we saw generated the most diverse sentences on a very small set of examples for a single domain and replicated the same prompts template for the other domains. This construction of prompts was inspired by [A] as the starting point of our prompt construction and [B] which performs mixup on the embedding space rather than in the text space like we do.
>
> [A] Sahu, G., Rodriguez, P., Laradji, I., Atighehchian, P., Vazquez, D., & Bahdanau, D. (2022, May). Data Augmentation for Intent Classification with Off-the-shelf Large Language Models. In Proceedings of the 4th Workshop on NLP for Conversational AI (pp. 47-57).
>
> [B] Kim, Y., Jeong, S., & Cho, K. (2021). Linda: Unsupervised learning to interpolate in natural language processing. arXiv preprint arXiv:2112.13969.

---

### Official Review · Reviewer_XgUZ · 2023-08-03

**Soundness:** 4

**Excitement:**

4: Strong: This paper deepens the understanding of some phenomenon or lowers the barriers to an existing research direction.

**Missing References:**

https://aclanthology.org/2023.findings-eacl.9/ - mentioned above already

**Paper Topic And Main Contributions:**

This paper introduces PromptMix where the authors propose generating borderline class samples using a LLM and then re-labeling them again leads to better learned student models trained on the augmented data. They select a subset of classes and k(=2) examples per class to create the instruction prompt. Then they instruct gpt-3.5-turbo to generate samples which contain mix of classes and get them re-labeled back using LLMs again to fix the ambiguous generation cases. This leads to better performance compared to other data augmentation techniques in few-shot & zero-shot settings.

**Questions For The Authors:**

One question would be do you think PromptMix would perform worse in cases of task like Hate Speech detection ?
Because it's already shown in this paper (https://aclanthology.org/2023.findings-eacl.9/) that for tasks like Hate Speech there is no concept of mixup like 70% non-hate + 30% hate. Any sentence which has >0% hate is hateful irrespective of whether it's majority or minority class.

So it would be interesting to see this kind of analysis in the paper as well or maybe cite & mention the same in Limitations section.

**Reasons To Accept:**

1) The paper is very detailed and well-written.
2) The idea of generating borderline examples using LLMs makes sense intuitively and also the authors show ablation as to why the re-labeling step is important.
3) They compare with lot of other related baselines which makes for a complete study on the task at hand.
4) Another interesting analysis, is adding class descriptions in zero-shot setting alone lead to improvements over other approaches.

**Reasons To Reject:**

1) In Section 3.1, it's mentioned that authors choose random c=4 classes. What is the performance variance when this "c" set is changed for B77 and TREC6 which has >4 classes ?
2) Another thing to note would be the performance of PromptMix inherently depends on the quality of questions / answers generated by the LLM. So a high-quality model is essential for the success of this task. An analysis on that would be nice i.e. which open-source LLMs would work with this approach ? or maybe find a LLM parameter count cutoff (say 7B) lower than that performance of PromptMix would start degrading ?
3)  Report the standard deviations for all the main results presented. Because in few-shot and LLM-based settings it's important to understand the stability of the results obtained.



*EDIT* - I've read the rebuttal and the authors have cleared out most of the rejection points through subsequent experiments. Keeping the rating as Strong Accept. Thanks for the work. Nice read !

**Reproducibility:**

4: Could mostly reproduce the results, but there may be some variation because of sample variance or minor variations in their interpretation of the protocol or method.

**Reviewer Confidence:**

5: Positive that my evaluation is correct. I read the paper very carefully and I am very familiar with related work.

---

> ### Author Rebuttal · Authors · 2023-08-28
>
> Dear Reviewer,
>
> Thank you for your review. We are pleased to know that you found our methodology intuitive and that you liked our analysis. Thank you for the relevant work suggestions and for asking great questions. Getting answers to your questions allowed us to gain some interesting insights. We would also like to mention that we will release our codebase upon acceptance for reproducibility.
>
> We now respond to your comments in detail:
>
> > *In Section 3.1, it's mentioned that authors choose random c=4 classes. What is the performance variance when this "c" set is changed for B77 and TREC6 which has >4 classes ?*
>
> Good question! We agree we didn't discuss our choice of $t=4$ in the paper, but we chose it based on the following validation accuracies obtain on the TREC6 for PromptMix (we have added these results to our paper):
>
> | t | Validation Acc (A2) |
> | --- | --- |
> | 2 |  73.3 |
> | 4 |  76.8 |
> | 6 |  69.1 |
> ||
>
> > *An analysis on that would be nice i.e. which open-source LLMs would work with this approach ? or maybe find a LLM parameter count cutoff (say 7B) lower than that performance of PromptMix would start degrading ?*
>
> Great question! We have tried the open-source models GPT-Neo (1.3B), GPT-J (6B), and the more recent instruction-tuned Vicuna and Stable Vicuna (13B) models but they achieved low performance in comparison to GPT-3.5.
>
> However, new open-source models have emerged recently in the LLM space, like Llama-2. As seen in the Table below, we ran different sized Llama-V2 (7b, 13b, and 70b) on TREC 6 and saw that Llama-V2 70b achieves close results to GPT-3.5, suggesting that current open-source models are also feasible for our proposed method. We have added these results to the paper.
>
>
> **Exp 1. Results on TREC 6**
>
> | Model |  Test Acc (A2) |
> | --- | --- |
> |**PromptMix Results**|
> | Llama-2-7b-chat-hf | 55.6$^\dagger$ |
> | Llama-2-13b-chat-hf |  66.6$^\dagger$ |
> | Llama-2-70b-chat-hf |  70.8$^\dagger$ |
> | GPT-3.5-turbo (as in the paper)  |  73.7 |
> |**Upper Bound Results**|
> |NN+GPT-3.5-turbo (as in the paper)  |  74.4 |
> ||
>
> $^\dagger$ used Llama-70b-chat-hf for relabeling.
>
> For the LLM parameter count cutoff, there's a significant performance drop when switching to the 7b Llama model, suggesting that 13B might be the minimum size for an LLM to be feasible for data augmentation in 2-shot text classification settings.
>
> > *Report the standard deviations for all the main results presented. Because in few-shot and LLM-based settings it's important to understand the stability of the results obtained.*
>
> Thanks for raising this point. We have added standard deviations to our main results table in parentheses. Here's a sample of Table 3 with std values (A1 and A2 values are separated by a ";").
>
> | Model | B77 | TREC6 | SUBJ | TC |
> | --- | --- | --- | --- | --- |
> | | | BERT-base | |
> | Baseline | 22.6 (1.2) | 33.0 (0.6) | 71.6 (0.8) | 42.7 (0.6) |
> | EDA | 49.2 (0.9) | 51.1 (0.6) | 84.5 (0.5) | 47.8 (0.4) |
> | GPT3Mix | $-$ | 60.5 (6.1) | 90.6 (1.1) | $-$ |
> | Linda (5-shot) | $-$ | 62.2 (3.1) | $-$ | $-$ |
> | Sahu et al., (2022) | 70.7 (1.4) ; 71.6 (1.1) | 51.3 (1.2) ; 57.8 (0.8) | 83.6 (1.3) ; 85.5 (1.0) | 55.3 (1.1); 58.1 (0.3) |
> | $\quad$+desc | 72.8 (1.1) ; 73.0 (0.7) | 64.3 (1.3) ; 67.1 (0.2) | 87.0 (1.7) ; 87.2 (1.1) | 66.1 (1.4) ; 65.2 (1.1) |
> | PromptMix w/o Mixup (zero-shot) | 74.0 (1.9) ; 76.4 (1.1) | 64.2 (1.6) ; 68.5 (0.9) | 83.6 (1.4) ; 85.9 (0.8) | 64.3 (1.2) ; 68.9 (0.3) |
> PromptMix (zero-shot) | 71.3 (2.4) ; 77.6 (1.3) | 55.5 (1.7) ; 67.5 (0.4) | 80.7 (1.4) ; 89.5 (0.7) | 57.6 (2.4) ; 74.7 (1.5) |
> PromptMix w/o Mixup | 74.4 (1.4) ; 78.5 (0.7) | 70.1 (1.5) ; 71.6 (0.2) | 87.0 (1.3) ; 90.0 (0.1) | 71.0 (0.9) ; 72.3 (0.3) |
> PromptMix | 74.2 (1.6) ; 80.1 (0.9) | 63.3 (2.5) ; 73.7 (1.1) | 77.3 (2.2) ; 91.7 (1.1) | 66.8 (0.8) ; 78.4 (0.8) |
> ||
>
> > *do you think PromptMix would perform worse in cases of task like Hate Speech detection ? Because it's already shown in this paper (https://aclanthology.org/2023.findings-eacl.9/) that for tasks like Hate Speech there is no concept of mixup like 70% non-hate + 30% hate. Any sentence which has >0% hate is hateful irrespective of whether it's majority or minority class.*
>
> No, we think PromptMix would work for Hate Speech Detection as well. The example of 70% non-hate + 30% hate should be correctly classified as hate, but it's possible that an ML model puts more attention onto the non-hate part of the sentence and misclassifies it as non-hate. In our pipeline, we aim to minimize such mistakes by training the classifier on such hard/ambiguous examples where examples are not clear for the model on whether they are "hate" or "non-hate". We also thank you for pointing out the EasyMix paper--we agree it carries the same motivation as our approach, and we will cite it and discuss it in our paper.

---

### Official Review · Reviewer_Z33a · 2023-08-05

**Soundness:** 4

**Excitement:**

4: Strong: This paper deepens the understanding of some phenomenon or lowers the barriers to an existing research direction.

**Missing References:**

Some recent work on distillation from LLMs to improve downstream tasks should be discussed in related work or introduction. See https://arxiv.org/abs/2110.07178 and follow-up works.


**Paper Topic And Main Contributions:**

This paper presents a data augmentation strategy for text classification using LLMs. The authors propose to use LLMs like GPT3 to generate "border-line" examples by (1) prompting the model to generate examples that are x% in one class and (1-x)% in another (mixing), and then (2) prompting the model again to label the generated examples. By augmenting these examples to the original training datasets and fine-tuning BERT based models, the authors show significant improvements over other data augmentation based baselines on the 4 datasets considered and improved performance over GPT3.5 on 3/4 datasets considered.

The authors also provided ablations on each of the proposed components to showcase the importance of each of them.

**Reasons To Accept:**

1. The proposed experimental setup is simple and easy to implement and leads to substantial improvements over baselines.
2. The paper is well written, well organized, and provides extensive analysis and ablations showcasing the importance of each presented component.

**Reasons To Reject:**

Not necessarily reasons to reject but the paper could benefit from including some of these discussions.
1. The reasoning behind why borderline examples are beneficial is not super clear from this draft. Some prior work has suggested that ambiguous examples are usually harder for models and having more of those usually helps improve model performance (see https://arxiv.org/abs/2009.10795).
2. One analysis that could strengthen the paper: which models are being used for data augmentation. How does the performance change with the size of the model considered for augmentation and so on?

**Reproducibility:**

4: Could mostly reproduce the results, but there may be some variation because of sample variance or minor variations in their interpretation of the protocol or method.

**Reviewer Confidence:**

3: Pretty sure, but there's a chance I missed something. Although I have a good feel for this area in general, I did not carefully check the paper's details, e.g., the math, experimental design, or novelty.

---

> ### Author Rebuttal · Authors · 2023-08-28
>
> Dear Reviewer,
>
> Thank you for your thoughtful feedback. We are encouraged to know that the paper is easy to follow and that you found our insights helpful. We thank you for the relevant citation suggestions and a meaningful analysis we can add to that paper. Overall, they have helped us gain new insights and discussion points. We would also like to mention that we will release our codebase upon acceptance for reproducibility.
>
> We now respond to your comments in detail:
>
> > *The reasoning behind why borderline examples are beneficial is not super clear from this draft. Some prior work has suggested that ambiguous examples are usually harder for models and having more of those usually helps improve model performance (see https://arxiv.org/abs/2009.10795).*
>
> Thank you very much for bringing this work to our attention! The concept of Data Maps highly aligns with our case-in-point that we need ambiguous examples to build a good quality dataset and ultimately train a robust classifier. We will include this work in our paper.
>
> > *One analysis that could strengthen the paper: which models are being used for data augmentation.*
>
> We use OpenAI's gpt-3.5-turbo for data augmentation (version 0613).
>
> > *How does the performance change with the size of the model considered for augmentation and so on?*
>
> Great question! We tried the smaller models GPT-Neo (1.3B), GPT-J (6B), OpenAI's Curie (estimated 13B), and the more recent instruction-tuned Vicuna and Stable Vicuna (13B) models but they achieved low performance in comparison to GPT-3.5.
>
> However, new models have emerged recently in the LLM space, like Llama-2 and GPT-4. As seen in the Table below, we ran different-sized Llama-2 models (7b, 13b, and 70b) on **TREC6** and saw a strong correlation between model size and test accuracy. We also ran GPT-4 (which is larger than the GPT-3.5-turbo model we used) and achieved even better results. We have added these results to the paper.
>
> **Exp 1. Results on TREC 6**
>
> | Model |  Test Acc (A2) |
> | --- | --- |
> |**PromptMix Results**|
> | Llama-2-7b-chat-hf | 55.6$^\dagger$ |
> | Llama-2-13b-chat-hf |  66.6$^\dagger$ |
> | Llama-2-70b-chat-hf |  70.8$^\dagger$ |
> | GPT-3.5-turbo (as in the paper)  |  73.7 |
> | GPT-4 |  **86.2*** |
> |**Upper Bound Results**|
> |NN+GPT-3.5-turbo (as in the paper) |  74.4 |
> |NN+Llama-2-70b-chat-hf | 76.2 |
> |NN+GPT4 | 88.2 |
> ||
>
> $^\dagger$ used Llama-2-70b-chat-hf for relabeling; *used GPT-4 for relabeling
>
> > *Some recent work on distillation from LLMs to improve downstream tasks should be discussed in related work or introduction. See https://arxiv.org/abs/2110.07178 and follow-up works.*
>
> Thank you for pointing this out. We will cite this work and add a discussion of these models in our related work section.

---

### Meta-Review · Area_Chair_duUG · 2023-09-17

**Recommendation:** 5

**Metareview:**

The paper proposes a data augmentation approach using pre-trained LLMs, where they generate borderline (challenging) examples using a model GPT3, then prompting the model again to re-label the examples for correcting potential labeling errors. They show experimentally that augmenting this kind of data leads to improved student models, including in challenging few and zero-shot settings. The paper is well written and organized, with extensive analyses and ablation studies supporting the claims of the paper, and demonstrating the validity of the proposed data augmentation approach across various classification tasks.

---

### Decision · Program_Chairs · 2023-10-07

**Decision:**

Accept-Main

**Comment:**

The paper proposes a data augmentation approach using pre-trained LLMs, where they generate borderline (challenging) examples using a model GPT3, then prompting the model again to re-label the examples for correcting potential labeling errors. They show experimentally that augmenting this kind of data leads to improved student models, including in challenging few and zero-shot settings. The paper is well written and organized, with extensive analyses and ablation studies supporting the claims of the paper, and demonstrating the validity of the proposed data augmentation approach across various classification tasks.